# Impact of Nutrition on Enhanced Recovery After Surgery (ERAS) in Gynecologic Oncology

**DOI:** 10.3390/nu11051088

**Published:** 2019-05-16

**Authors:** Steven Bisch, Gregg Nelson, Alon Altman

**Affiliations:** 1Division of Gynecologic Oncology, Tom Baker Cancer Centre, Calgary, AB T2N4N2, Canada; steven.bisch@ahs.ca; 2Department of Obstetrics, Gynecology and Reproductive Sciences, University of Manitoba, Winnipeg, MB R3A 1R9, Canada; alondaltman@gmail.com

**Keywords:** ERAS, enhanced recovery after surgery, gynecologic/oncology, nutrition

## Abstract

Enhanced recovery after surgery (ERAS) pathways aim to improve surgical outcomes by applying evidence-based practices before, during, and after surgery. Patients undergoing surgery for gynecologic malignancies are at high risk of complications due to population, patient, disease, and surgical factors. The nutritional status of the patient provides the foundation for recovery after surgery, and opportunities to optimize outcomes exist from the first patient assessment to the early days after surgery. This review highlights the importance of nutritional assessment and intervention during the pre-operative and post-operative periods in the context of ERAS in gynecologic oncology surgery. The emerging role of immunonutrition, carbohydrate loading, and the importance of individualized care are explored. Evidence from studies in gynecologic oncology is presented, where available, and extrapolated from colorectal and other cancer surgery trials when applicable.

## 1. Introduction

Gynecologic malignancies are a diverse set of diseases that affect women from all demographic and cultural backgrounds. The heterogeneity of gynecologic malignancies presents a unique nutritional challenge; ovarian cancer patients often present at an advanced stage, and up to 20% are clinically malnourished [1]. Conversely, endometrial cancer is often diagnosed at an early stage and is associated with elevated body mass index (BMI) and insulin resistance [2,3]. Within cervical cancer patients, rates of malnourishment vary considerably; up to 60% of patients with stage IV disease are malnourished while this rate is only 4% in those with stage I disease [1]. Further complicating matters is the nature of treatment regarding gynecologic malignancies; radical surgery is often a necessary element in the treatment of advanced disease [4]. Nutritional status is a marker of perioperative morbidity and mortality [4], and obesity and insulin resistance are independent risk factors for perioperative morbidity [5]. The diverse demographics and disease-related differences of this population create a challenge for clinicians caring for patients in the perioperative period.

This review examines the major sections in perioperative nutritional care as it relates to gynecologic oncology and enhanced recovery after surgery (ERAS) recommendations and explores patient populations where treatment should be individualized. A summary of these findings, with references, is presented in Table 1.

## 2. Pre-Operative Nutritional Status

Pre-operative malnutrition and serum albumin levels are known to be associated with increased morbidity and mortality, [13] and improving pre-operative nutrition for 7–14 days prior to surgery improves outcomes [11,12,13].

Various nutritional screening scores have been used to help identify patients at risk, including the following: nutritional risk screening score (NRS 2002) [6], subjective global assessment (SGA) [7], patient-generated subjective global assessment (PG-SGA) [8], malnutrition universal screening tool (MUST) [9], and preoperative nutrition screen (PONS) [10]. Although, the various scoring systems have not been directly compared, all use the same basic structure and are easy to implement in clinical settings. Other studies have defined severe nutritional deficiency as per the European Society for Clinical Nutrition and Metabolism (ESPEN) guidelines: Weight loss > 10–15% in past 6 months, BMI < 18.5 kg/m^2^, SGA = C, NRS > 5, or albumin < 30 g/L [13]. Patients with nutrition that is expected to be less than 50% of their daily dietary total should be optimized pre-operatively and continue post-operatively with preferably enteral feeding. Parenteral nutrition can be used when enteral means do not meet energy needs [13]. Supplementation for 10 days pre-operatively has been found to decrease complications (prolonged air leak and atelectasis requiring bronchoscopy) in patients undergoing surgery for lung cancer [41].

The discontinuation of excessive alcohol intake 4–6 weeks pre-operatively may be associated with a decrease in post-operative complications, but further research with high-quality studies is needed [14].

Screening and correcting malnutrition in the pre-operative period is a key element in enhanced recovery pathways by facilitating an optimal environment to handle the impact of surgery, especially in the presence of gynecologic malignancies.

## 3. Pre-Operative Fasting

The initial concept of pre-operative fasting was brought about as a preventative strategy for regurgitation and aspiration. The first recorded aspiration pneumonitis following anesthesia was reported in 1848. One of the first papers outlining this concept was published by Mendelson (1946) [42] and describes the incidence of aspiration in the obstetric population. Although only two patients died in this series, it became the cornerstone of anesthetic policy and the drive for “nothing by mouth” for all pre-operative patients [17,42], with a goal of less than 200 mL of fluid in the stomach prior to anesthetic interventions. Pre-operative fasting policies are not without harm; prolonged states of fasting lead to increased insulin resistance, hyperglycemia, catabolic metabolism, and muscle breakdown [43].

Even though the actual risk of pulmonary aspiration remains low [44,45], the practice of “nil per os (NPO)” has continued for decades and has only recently been challenged. Pre-operative fasting unfortunately does not take into account electrolyte imbalance, dehydration, and insulin resistance [17]. Many studies and reviews have now compared perioperative stress with high-performance athletic activities, as the two events have similar demands regarding nutritional requirement, anaerobic metabolism, and oxygen requirements [17,46]. Pre-operative fasting leads to a fasted state, thereby reducing liver glycogen stores, increasing insulin resistance, and increasing the post-operative stress response. Post-operative hyperglycemia is independently associated with increased morbidity and mortality [27,47,48,49].

Numerous studies have now shown that a clear fluid diet up to 2 h pre-operatively does not increase gastric content, reduce the pH of the gastric fluid when compared with fasting, or increase complication rates [15]. Gastroparesis, peritonitis, and conditions associated with delayed gastric emptying have not been well examined and require further investigation. Care should be paid to situations with functional dyspepsia, increased estrogen/progesterone levels, autonomic neuropathy, and states of increased stress until more research is performed. An increase in circulating estrogen has been demonstrated to inhibit gastric emptying [26], and care should be taken in patients with estrogen-secreting tumors (e.g., adult granulosa cell ovarian tumors). With the caveat of the aforementioned special instances, the intake of clear fluids until 2 h before surgery and a six-hour fast for solid food is now recommended by most anesthetic guidelines [13,16,17,18].

## 4. Carbohydrate Preloading

Carbohydrate loading in athletes has been shown to improve performance and increase muscle glycogen stores [50,51,52]. A “fasting state” can be triggered as quickly as 4 h after the last meal [43], which can lead to post-operative insulin resistance and hyperglycemia; this can therefore increase risks for post-operative complications, including increased length of stay, renal failure, infection, reoperation, myocardial infarction, and death [43,48,53]. Starvation is postulated to cause insulin resistance through mitochondrial dysfunction [54]. Carbohydrate loading before surgery has been advocated to achieve reduced inflammation, increased insulin sensitivity, improve post-operative muscle function, and improve patient reported outcomes [25,46,55]. The pre-operative carbohydrate load appears to have little effect on gastric emptying [13]. Recently, an increasing number of studies, systematic reviews, and meta-analyses on this issue in colorectal surgery have been published. They show that carbohydrate loading improves insulin resistance related to the surgical procedure, decreases the length of hospitalization, and should be used routinely [17,19,20,21,22,23,24,56]. Moreover, carbohydrate loading is associated with significant improvements in patient wellbeing, with decreased hunger and anxiety being reported by patients [25]. Although no studies have been performed in patients undergoing major gynecological or gynecologic oncology surgery, these findings are still likely valid in this patient population. Many gynecologic oncology procedures include extensive dissection and resection, causing large surgical and physiologic stress on the body.

The most commonly used oral supplement is maltodextrin (reconstituted in a clear fluid) in a 50 g packet with a 100 g evening dose and a 50 g morning dose ~3 h pre-operatively. The evening dose of carbohydrate is meant to create glycogen stores, whereas the morning dose is meant to change the body to a “fed” state [46,57]. To be effective, the pre-operative carbohydrate load needs to be ~12% carbohydrate with an osmolality of 135 mOsm/kg [46], where most “sports” drinks are approximately 6–7%. Carbohydrate polymers in oral supplements act to minimize osmotic load and reduce gastric emptying time [27]. Carbohydrate loading from other dietary sources (e.g., pasta) or supplements (e.g., exercise gels) have not yet been studied.

Very few studies have been identified that specifically address diabetic patients. Gustafsson et al. (2008) examined 25 type 2 diabetic patients compared with 10 non-diabetic patients. Although there was a difference in time to peak glucose concentration, there were no negative changes in gastric emptying [27]. Although diabetic patients are at risk of delayed gastric emptying, they are also the most likely to benefit from the reduction in hyperglycemia afforded by avoidance of fasting and pre-operative carbohydrate intake. Modern fasting guidelines may be safely used for patients with uncomplicated type 2 diabetes; however care should be individualized for patients with type 1 diabetes or longstanding poorly controlled diabetes [27].

Implementation of pre-operative carbohydrate loading is currently limited due to resistance from clinicians (physicians, anesthesiologists, and nurses), logistical challenges, lack of funding, and pharmacy issues [17]. A multidisciplinary approach with consistent and clear information is essential at all stages of pre-operative patient education to ensure appropriate guideline adherence and potential surgical delay.

## 5. Post-Operative Nutritional Care

Fears around complications associated with feeding have led to historical avoidance of food in the post-operative period. Patients have traditionally been advised to take nothing by mouth, with a slow progressive diet through clear fluids, full fluids, and solid diet. Although recent strides have been made to combat this surgical myth, change has been slow, leading to increased nutritional issues in patients, slowing recovery and worsening outcomes.

The maintenance of an appropriate nutritional status in the post-operative period is recommended and supported [10]. Multiple randomized studies on early enteral feeding have been performed in gynecologic oncology surgery [32,33,34,35,36,58], with most studies aiming to provide caloric intake by mouth within 24 h post-operatively. Studies in gynecologic oncology surgery have demonstrated reduced length of stay, as well as improved time to first flatus, without increased rates of complications. Most studies have demonstrated equivalent or even improved complication rates, with no differences in anastomotic leaks, wound healing, or pulmonary complications [35,37,58]. A consequence of early feeding is a higher rate of nausea, but with modern post-operative nausea and vomiting regimens, there has been no increase in vomiting or nasogastric tube insertion. In colorectal patients, delivery of post-operative nutrition on day one is an independent prognostic factor for five-year survival and mortality [10,59,60]. Many gynecologic oncology centers have progressed to allow their patients a standard diet immediately post-operation, with no apparent increase in rates of complications [13]; however, uptake is far from universal. The exact composition of a post-operative diet is up for debate; however, a high protein diet post-operatively may reduce the rate of complications [10]. Currently surgical patients do not have clear guidelines on protein needs; however, acute care guidelines have recommended 2.0 g of protein/kg/day and 25–30 kcal/kg/day [13,61]. These nutrition levels are further supported by the American Society for Enhanced Recovery and Perioperative Quality guidelines [10].

## 6. Immunonutrition

Perioperative nutritional supplementation and immune nutrition are a rapidly expanding area of research and focus. Immunonutrients are those nutrients that have demonstrated a measurable effect on the immune system through their supplementation [30]. Current research aims to attenuate the effect of post-operative inflammation as well as improve post-operative healing. Nutrients of interest in current research include the following: polyunsaturated/omega-3 fatty acids, arginine, glutamine, antioxidants, and nucleotides [10,13,17,28,62]. A large systematic review of 35 randomized controlled trials showed a reduction in overall infection (RR = 0.59) and length of hospital stay with the supplementation of arginine in perioperative diet, with no differences in mortality [63]. Several large randomized trials in colorectal patients compared immune nutrition/high-protein feeds to a high-calorie supplement and found a lower rate of infection and length of stay in the study group [28,29]. Although many of the trials examining immunonutrition involve bowel or colorectal surgery, a recent randomized controlled pilot study of urology patients undergoing radical cystectomy demonstrated an attenuated inflammatory response in individuals receiving perioperative immunonutrition compared with the controls [30]. One study in gynecologic oncology by Celik et al. (2009) randomized 50 patients to receive immune-enhancing versus standard enteral nutrition; patients in the immune nutrition group tolerated the diet well, with increased evidence of immunologic responses and shorter hospital stays [31].

Immunonutrition may be most beneficial in the pre-operative period and may be more important than post-operative immunonutrition in all but the most complex surgeries. A recent study in patients undergoing pancreaticoduodenectomy found that pre-operative immunonutrition alone was equivalent to both pre- and post-operative supplementation in attenuating the negative immune responses of stressful surgery [40]. A recent study of ERAS in gynecologic oncology patients demonstrated compliance with oral nutritional supplements of only 50%. This was thought to be due to a rapid progression to full diet and less patient reliance on oral supplementation [37].

## 7. Gum Chewing

Gum chewing has also been used in many ERAS protocols and by many centers to enhance post-operative recovery. Chewing gum acts as a form of sham feeding and may reduce post-operative ileus by stimulation of the gastrointestinal system in the absence of feeding [39]. The hexitols in sugar-free gum may also act osmotically to reduce the risk of ileus [64]. In 2015, a Cochrane review by Short et al. examined 81 studies with 9072 patients. Gum was found to reduce the time to first flatus and time to first bowel movement in both colorectal and caesarean section patients. There was also statistical evidence that gum slightly reduced length of hospital stay [38]. In contrast, a more recent randomized control trial showed no difference in length of stay or rate of ileus [39]. The benefits of chewing gum may also be diminished in the context of other interventions in ERAS pathways [39]. Gum chewing may not be tolerated by patients due to physical (dentures) or personal reasons (appearance of chewing gum, or social norms against speaking while chewing gum) in as much as one-third of patients [65]. Current use of gum remains somewhat controversial; however, it has relatively low risk complications and is still used by many ERAS centers.

## 8. Conclusions

Perioperative nutrition management is a crucial component of any ERAS protocol. Examination of the patient should begin prior to surgery, with appropriate clinical assessment and screening for malnutrition. Malnourished patients should receive optimal nutritional supplementation, beginning with enteral feeds and using parenteral methods if necessary. Ideally, treatment should begin 7–14 days pre-operatively. Avoiding a fasting state prior to surgery can be achieved by limiting fasting times and carbohydrate loading both the night before surgery and 3 h pre-operatively. Post-operative nutrition should be started immediately after surgery and can be advanced rapidly. Nutritional management should be tailored to specific patient needs and comorbidities using evidence-based decisions whenever possible. Multiple benefits from aggressive nutritional treatment can be achieved with a collaborative and multi-disciplinary approach, harnessing physicians, anesthesiologists, nurses, and dieticians.

## Figures and Tables

**Table 1 nutrients-11-01088-t001:** Summary of Considerations.

Time	Summary	References
Early Pre-op	Pre-operative nutritional assessment with validated scoring tool	Kondrup, 2003 [6]; Detsky, 1987 [7]; Ottery, 1994 [8]; Stratton, 2004 [9]; Wischmeyer, 2018 [10]
	Supplement malnourished patients 10–14 days pre-op	Bozzetti, 2007 [11]; Jie, 2012 [12]; Weimann, 2017 [13].
	Alcohol cessation 4–6 weeks pre-op	Egholm, 2018 [14]
Late Pre-op	Intake solid foods 6–8 h pre-opIntake clear fluids 2–3 h pre-op	Brady, 2003 [15]; Smith, 2011 [16]; Scott, 2014 [17]; Merchant, 2016 [18]
	Carbohydrate loading drink (50 g maltodextrin) ~3 h pre-op	Weimann, 2017 [13]; Yagci, 2008 [19]; Awad, 2013 43]; Gustafsson, 2013 [20]; Bilku, 2014 [21]; Ljunggren, 2014 [22]; Scott, 2014 [17]; Smith, 2014 [23]; Webster, 2014 [24]; Hausel, 2001 [25]
	Individualize care in patients with risk of delayed gastric emptying or diabetes	Chen, 1995 [26]; Gustafsson, 2008 [27]
	Pre-operative immunonutrition may be able to alter the immune response to surgery	Moya, 2016 [28]; Yeung, 2017 [29]; Hamilton-Reeves, 2018 [30]; Celik, 2009 [31]
Post-op	Nutritional support (ideally enteral feeding) starting within 24 h post-operatively Consider increased protein and caloric demands of cancer and recent surgery	Schilder, 1997 [32]; Pearl, 1998 [33]; Cutillo, 1999 [34]; Minig, 2009 [35]; Charoenkwan, 2014 [36]; Bisch, 2018 [37]
	Benefits of chewing gum for stimulation of gut motility are controversial	Short, 2015 [38]; Atkinson, 2016 [39]
	Post-operative immunonutrition may not be as important as pre-operative supplementation—further research is needed.	Miyauchi, 2019 [40]

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
