# Peer review of "Impact of Nutrition on Enhanced Recovery After Surgery (ERAS) in Gynecologic Oncology"

_nutrients, 2019, doi:10.3390/nu11051088_

Round 1

Reviewer 1 Report

In this review, the authors discuss the importance of nutritional assessment and intervention during the pre-operative and post-operative period in the context of ERAS (Enhanced Recovery After Surgery) in gynecologic oncology surgery.

This is a well-organized review, but I have some comments.

The authors should provide an explanation for ‘immunonutrition.’ Why are these nutrients (polyunsaturated/omega-3 fatty acids, arginine, glutamine, antioxidants, and nucleotides) immune-enhancing? 

In the ‘Preoperative nutritional status’ section, five nutritional screening scores are listed. References for these scores are necessary.

Line 56. The discontinuation of excessive alcohol intake. 4-6 weeks (or days?) preoperatively?

Line 83. Although I can imagine what ‘gastrostasis’ is, the authors should give the definition of ‘gastrostasis.’

Line 84. Why should care be paid to the situation with increased estrogen/progesterone levels? In gynecologic oncology, what diseases are associated with this situation? Please explain.

Line 107. Maltodextrin is a purified concentrated aqueous solution or the dried product from the solution. In order to avoid confusion, please clarify.

References. Volumes and pages are missing in many citations.  

Author Response

The authors should provide an explanation for ‘immunonutrition.’ Why are these nutrients (polyunsaturated/omega-3 fatty acids, arginine, glutamine, antioxidants, and nucleotides) immune-enhancing? 

Completed

In the ‘Preoperative nutritional status’ section, five nutritional screening scores are listed. References for these scores are necessary.

Completed 

Line 56. The discontinuation of excessive alcohol intake. 4-6 weeks (or days?) preoperatively?

Completed

Line 83. Although I can imagine what ‘gastrostasis’ is, the authors should give the definition of ‘gastrostasis.’

Completed

Line 84. Why should care be paid to the situation with increased estrogen/progesterone levels? In gynecologic oncology, what diseases are associated with this situation? Please explain.

Completed

Line 107. Maltodextrin is a purified concentrated aqueous solution or the dried product from the solution. In order to avoid confusion, please clarify.

Completed

References. Volumes and pages are missing in many citations. 

Completed

Reviewer 2 Report

well written review

Making a summary table may be helpful to guide the nutritional support or ERAS for gynecologist or gynecologic oncologist. 

Author Response

Making a summary table may be helpful to guide the nutritional support or ERAS for gynecologist or gynecologic oncologist. 

Completed (see attached new Table 1)